# Clinical Relevance of Ultrasonographic and Electrophysiological Findings of the Median Nerve in Unilateral Carpal Tunnel Syndrome Patients

**DOI:** 10.3390/diagnostics12112799

**Published:** 2022-11-15

**Authors:** Takamasa Kudo, Yuichi Yoshii, Yuki Hara, Takeshi Ogawa, Tomoo Ishii

**Affiliations:** 1Department of Orthopedic Surgery, Tokyo Medical University Ibaraki Medical Center, Ami 300-0395, Ibaraki, Japan; 2Department of Orthopedic Surgery, University of Tsukuba Hospital, Tsukuba 305-8577, Ibaraki, Japan; 3Department of Orthopaedic Surgery, Mito Medical Center, Mito 311-3193, Ibaraki, Japan

**Keywords:** carpal tunnel syndrome, nerve conduction study, ultrasound, median nerve

## Abstract

Few studies have compared the unaffected and affected sides in the same carpal tunnel syndrome (CTS) patients using ultrasonography and electrophysiological tests. We focused on unilateral idiopathic CTS patients to investigate whether clinical test results differ between the unaffected and affected sides. The bilateral wrist joints of 61 unilateral idiopathic CTS patients were evaluated. The median nerve cross-sectional area of ultrasound image, and latencies of the compound muscle action potential (CMAP) and sensory nerve action potential (SNAP) were measured. The values obtained were compared between the affected and unaffected sides. The diagnostic accuracies of each parameter were assessed, and cut-off values were defined. Significant differences were observed in all parameters between the affected and unaffected sides (*p* < 0.01). Area under the curve (AUC) values were 0.74, 0.88, and 0.73 for the cross-sectional area, CMAP distal latency, and SNAP distal latency, respectively. Cut-off values were 11.9 mm^2^, 5.1 ms, and 3.1 ms for the cross-sectional area, CMAP distal latency, and SNAP distal latency, respectively. The most reliable parameter that reflected clinical symptoms was the distal latency of CMAP. Cut-off values for each parameter are considered to be an index for the onset of the clinical symptoms of CTS.

## 1. Introduction

Carpal tunnel syndrome (CTS) is defined as a compression neuropathy of the median nerve at the wrist joint. As clinical findings, night pain has been identified as a sensitive symptom predictor (96%), and the useful signs for the diagnosis are sensory impairment in the median nerve area of the hand (76%) and Tinel’s sign (71%) [1]. However, clinical findings are subjective as they are based on self-reported symptoms by patients. In addition, symptoms may occur outside of the median nerve control area [2]. Therefore, difficulties are associated with reliably diagnosing CTS based solely on clinical findings. CTS is commonly diagnosed by a comprehensive assessment of clinical signs, electrophysiology, and imaging. Many studies have reported the utility of electrophysiological tests and ultrasonography for comparing CTS patients with normal subjects [3,4]. However, few studies have compared the unaffected and affected sides in the same CTS patients because many patients have bilateral symptoms. Therefore, we herein focused on unilateral idiopathic CTS patients and investigated whether clinical test results differed between the unaffected and affected sides. The aim of the present study was to characterize the findings of ultrasonography and nerve conduction studies on the symptomatic and asymptomatic sides of unilateral CTS patients and to indicate the cutoff values for the symptom expression. This study attempted to identify the clinical parameters in which patients perceive their symptoms by comparing the results of clinical examinations of symptomatic and asymptomatic hands. We hypothesized that morphology and nerve conduction of the median nerve differ between the symptomatic and asymptomatic sides in the CTS patients with unilateral symptom.

## 2. Materials and Methods

The protocol for the present study was reviewed and approved by our Institutional Review Board (approved number T2020-0061). The bilateral wrist joints of 61 unilateral idiopathic CTS patients (122 wrists, 22 males, 39 females, 30–89 years, mean 65.3 years) were evaluated. Patients with chronic kidney disease, thyroid disease, and rheumatoid arthritis were excluded. Patients with a history of upper limb surgery were also excluded. CTS was diagnosed based on clinical symptoms and the results of motor and sensory nerve conduction studies as well as ultrasonography. Patients were asked to complete the JSSH version of the CTS instrument [5]. Clinical evaluation included the presence of typical sensory symptoms, Phalen’s test and Tinel’s sign, sensory testing by two-point discrimination on the middle finger, muscle testing, and examination for thenar atrophy. Informed consent was obtained from all patients for inclusion in the present study. In this study, we defined unilateral CTS as patients with characteristic symptoms on one side of the hand and no symptoms on the other side. The diagnosis of CTS was confirmed based on the clinical symptoms of pain, numbness, tingling in the median nerve distribution of the hand, and the presence of at least one positive provocative test result, in addition to meeting the criteria of nerve conduction study findings in the American Association of Electrodiagnostic Medicine (AAEM) guidelines [6]. For the asymptomatic side, the patients were also asked about the clinical symptoms of pain, numbness, and muscle weakness. Then, the strength of APB muscle and positivity of provocation tests were evaluated. It was defined as asymptomatic if none of the above clinical symptoms and evaluations were observed. A single hand surgeon discriminated between the asymptomatic and symptomatic sides based on interview and clinical findings. All patients underwent a nerve conduction study and ultrasound imaging. 

The cross-sectional area of the median nerve was measured by following method. Ultrasound imaging was routinely performed in patients with suspected CTS to differentiate abnormalities around the carpal tunnel. During the diagnostic process, the cross-sectional area of the median nerve was measured at the wrist crease level (proximal carpal tunnel) (Figure 1). Each patient was asked to sit and place their forearm on the table with the palmar side up. An ultrasound scanner (Hi Vision Avius; Hitachi Aloka Medical, Ltd., Tokyo, Japan) equipped with a linear array transducer was set to a depth of 20 mm. The frequency of the transducer was 15 MHz. Cross-sectional ultrasonographic images of the carpal tunnel were analyzed using ImageJ Software (National Institutes of Health, Bethesda, MD, USA). The median nerve was outlined, and its area was calculated. All ultrasound studies were performed by a hand surgery specialist. The hand surgeon has been certified as a specialist and an instructor by the Japanese Society of Surgery of the Hand. 

The nerve conduction study was performed on all patients using a standard electromyography system (Neuropack MEB-2208, Nihon Kohden Co., Tokyo, Japan). All studies were performed by a clinical technician who was blinded to clinical symptoms. At the time of the nerve conduction study, room temperature was maintained at 27 °C. Among patients with cold hands, the hands were warmed to bring the skin temperature closer to room temperature. In the motor conduction study, the compound muscle action potential (CMAP) of the abductor pollicis brevis muscle was recorded. CMAP was induced by a stimulation 7 cm proximal to the recording electrode. In the sensory conduction study, a stimulating electrode was placed at the index finger, and a recording electrode was placed at 14 cm proximal to the stimulating electrode. The sensory nerve action potential (SNAP) was recorded. The latencies of CMAP and SNAP were measured (Figure 2). Results were excluded from the analysis if there was no action potential in some cases.

Statistical analysis was evaluated by the method described here. Results are expressed as the mean ± standard deviation. Measured values were compared between the affected and unaffected sides using the χ^2^ test. A receiver operating characteristic (ROC) curve was created using a logistic analysis. Diagnostic accuracy was evaluated by the area under the curve (AUC value) based on the ROC curve analysis. Cut-off values and sensitivity/specificity were assessed using the Youden Index method. All analyses were performed using Bellcurve for Excel (version 2.14).

## 3. Results

Table 1 shows the patient demographics. Table 2 shows the results obtained for each parameter. Figure 3 shows the ROC curves for each parameter. The cross-sectional areas of the median nerve in six cases were larger on the unaffected side than on the affected side. In the motor nerve conduction study, CMAP was not derived on the affected side in 12 cases. In the sensory nerve conduction study, SNAP was not derived on the affected side in 16 cases. CMAP and SNAP waveforms were derived on the unaffected side in all cases. Significant differences were observed in all parameters between the affected and unaffected sides (*p* < 0.01). AUC values were 0.74, 0.88, and 0.73 for the cross-sectional area, CMAP distal latency, and SNAP distal latency, respectively. Cut-off values were 11.9 mm^2^, 5.1 ms, and 3.1 ms for the cross-sectional area, CMAP distal latency, and SNAP distal latency, respectively. 

## 4. Discussion

In the present study, the cut-off values were defined in the cross-sectional areas and nerve conduction study distinguished CTS as symptomatic and asymptomatic. An important point in this study was that it could characterize asymptomatic conditions despite positive ultrasound and electrophysiological findings. This could be characterized by examining unilateral CTS patients. The most accurate diagnostic parameter in this study was the distal latency of CMAP. It has been known that most of CTS patients have bilateral symptoms. The asymptomatic side may develop symptoms in the future. By characterizing a unilateral asymptomatic situation in this study, it may allow early therapeutic intervention to prevent progression of neuropathy.

The diagnostic accuracy of ultrasonography was previously reported to be similar to that of the nerve conduction study [7]. Although ultrasonography alone has sufficient diagnostic accuracy, it has been pointed out that combining ultrasonography and nerve conduction study improves diagnostic accuracy [8]. Ultrasonography is less stressful on the patient and can identify neuromas or space-occupying lesions within the carpal tunnel, which could not be evaluated electrophysiologically. It can also identify a well-developed median artery and the compression of the median nerve associated with that arterial thrombus. In Nakamichi’s studies, space-occupying lesions were found only in the unilateral group (7/20 = 35%), and it is said that careful examination of unilateral carpal tunnel syndrome on suspicion of local pathology, especially space-occupying lesion, is important [9]. Prior confirmation of these findings is useful in selecting a treatment. However, device/examiner dependency on the visualization of peripheral nerves is a limitation of this technique [10,11]. Furthermore, swelling of the median nerve is difficult to detect in patients older than 80 years, regardless of their severity of CTS [12]. In addition, diabetic conditions are known to affect the size of the median nerve [13,14]. These factors should be considered when assessing median nerve size.

The present study also showed that the cross-sectional area of the median nerve was significantly higher on the affected side than on the unaffected side. In the present study, the cut-off value for the cross-sectional area was 11.9 mm^2^ and the AUC value was 0.74 (sensitivity: 72%, specificity: 68%). Cut-off values in previous studies varied between 6.5 and 15 mm^2^ [15,16,17]. Only three studies defined a cut-off value of 11.9 mm^2^ or higher. It has been pointed out that the cutoff value of cross-sectional area changes as conditions change [18], but in this study, differences within patients are compared, so the conditions are considered to be relatively homogeneous. Similar to the nerve conduction study, cut-off values were higher in the present study than that in a study comparing CTS patients with normal subjects. In addition, the asymptomatic hands of 62–85% of cases met the general diagnostic criteria (cross-sectional area 8.5–10 mm^2^ or higher) [19]. When considered in the context of the Japanese people, Sugimoto’s studies have shown that the average of Japanese median nerve cross-sectional area (carpal tunnel inlet at the pisiform bone level) was 8.5 ± 1.7 mm^2^, which is a study of healthy subjects [20]. It has been suggested that the physical characteristics such as sex, dominant hand, age, height, weight, body mass index (BMI) and wrist circumference are associated with nerve size [20]. Compared to Sugimoto’s study, the present case is older, shorter in height and BMI (the average ages of the subject were 35 and 65 years, heights were 164 cm and 157 cm, and BMI were 22.3 and 24, in the Sugimoto’s and the present studies, respectively). In that study, age and BMI were positively, and height was negatively correlated with cross-sectional area. Although it is pointed out that these factors may cause the cross-sectional area in this case to be larger, we think that the value of cross-sectional area was larger even taking these factors into account. These findings suggest that even patients with unilateral symptoms have morphological changes in the median nerve of the asymptomatic hand. 

Additionally, in the previous study, the mean median nerve cross-sectional area for carpal tunnel inlet seems to be similar for middle east (8.77 mm^2^), Oceania (8.71 mm^2^), Europe (8.90 mm^2^) and Asia (8.68 mm^2^) studies [21]. Although these were not directly compared, it shows that there is no racial difference in the cross-sectional area of median nerve. Despite the lack of significant differences between ethnic groups, the cutoff values for the present cases are larger compared to the general values, which based on values from previous literature [19]. One reason is that, as mentioned above, morphological changes in the median nerve is seen because the subjects of this study are unilateral symptoms. However, Nakamichi’s study compared the Japanese with normal subjects, and even then, the cutoff value for cross-sectional area was set at 12 mm^2^ (sensitivity 67%, specificity 97%), which is a higher value compared to the results of other countries [22]. This suggests that the cutoff value may be larger in the Japanese. From this finding, Japanese may be less likely to complain about physical symptoms than the patients in other country even when the swelling of the median nerve is present. 

Because ultrasound can easily observe changes over time, it may help to predict symptom onset or determine the therapeutic effect. For more accurate diagnosis, we are thinking to use the wrist to forearm ratio for the ultrasound parameter [23]. Ultrasound can also detect increased intraneural blood flow of the median nerve with using Doppler sonography. It has been pointed out that it may be an indicator of early diagnosis and severity [24]. Nerve compression caused by elevated pressure in the carpal tunnel is believed to provoke a three-stage process that is initiated with venous congestion of the median nerve followed by nerve edema and then by impairment of the venous and arterial blood supplies. Comparison of findings of sonography and nerve conduction studies showed that nerve hypervascularization and nerve swelling yielded the best detectability of carpal tunnel syndrome [25]. Because there were several methods to identify median nerve pathological changes, it is clear that cross-sectional area measurement alone is not sufficient for a comprehensive ultrasound evaluation of peripheral nerves [26]. It may be possible to improve the accuracy of diagnosis by combining several parameters.

The nerve conduction study is one of a clinical test that is used to diagnose CTS [6]. It also helps to distinguish CTS from other clinical disorders. Additionally, it has been pointed out that the severity of CTS can be evaluated. Furthermore, the effects of surgery may be objectively evaluated by performing the study before and after the surgery [27]. On the other hand, the nerve conduction study is affected by age, height, finger circumference, sex, and skin temperature [28,29,30,31,32]. Therefore, previous studies adjusted for backgrounds, such as age/sex and finger usage, to eliminate selection bias in comparisons of healthy subjects and CTS patients. In the present study, it was possible to define cut-off values for clinical tests in patients with CTS symptoms by comparing the symptomatic and asymptomatic sides in the same patients. We considered two reasons why the distal latency of CMAP was found to be the most accurate. First, it has been pointed out that, Japanese may be less likely to complain about physical symptoms than the patients in other country even when neurological abnormalities are present [33]. The average latency of CMAP in asymptomatic hands in this study was 4.4 ± 0.9 ms, suggesting that the condition progresses to some extent insidiously according to previous grading [34]. Generally, sensory nerves are damaged first, and the onset of CTS is confirmed by a decrease in the sensory conduction velocity. In the case of Japanese patients, it is thought that the first hospital visit is often made after the symptoms have progressed to the extent that CMAP is impaired. Second, SNAP is affected by the factors such as body surface and room temperatures, and often becomes unrecordable when the CTS symptoms become more severe [35]. Because of these factors, the distal latency of CMAP was found to be the most accurate in the present study.

Individuals with certain occupations, such as postal staff, healthcare professionals, builders, and assembly workers, are more susceptible to CTS [36]. Evaluations of electrophysiological tests revealed the following prevalence of carpal tunnel syndrome: 20% for forest workers using vibration tools, 23% for staff working in general merchandise stores, 53% for meat workers, 17.8% for furniture makers, and 30% for dentists [37,38,39,40]. A common factor in these occupations is repetitive compression of the median nerve. Abnormal nerve conduction has been detected in some clinically asymptomatic nerves and has been defined as subclinical CTS [41]. MCV decreases with age, even in healthy subjects [42,43]. Therefore, the asymptomatic side in unilateral CTS patients is considered to be in a state of subclinical CTS. A previous study reported that the rate of subclinical CTS was approximately 18% [33]. Cut-off values were higher in the present study than in a previous study that compared normal subjects and CTS patients. This difference was attributed to the values measured being higher in the asymptomatic hands of unilateral CTS patients than in normal subjects. Among asymptomatic hands, there were cases in which 34% of sensory nerve conduction study and 50% of motor nerve conduction study met the diagnostic criteria of carpal tunnel syndrome [7]. To improve diagnostic accuracy, it may be necessary to devise diagnostic criteria that combine ultrasonography and nerve conduction studies [17].

Since symptoms such as numbness and pain are subjective, they are difficult to quantify and adapt to all patients. Furthermore, these symptoms may be hidden by patients. This study compared values in symptomatic and asymptomatic hands in Japanese patients. Therefore, the cut-off values obtained are close to the values at which symptoms reach a level that interferes with daily activity. A previous study reported that approximately 30% of CTS patients showed significant improvements in their natural course [44]. Based on the values measured in the present study, patients with values near the cut-off values may be able to improve with conservative treatment. In addition, these values may be used as reference values for the selection of surgical treatment for patients. 

There were several limitations that need to be addressed. First, sex, dominant hand, age, height, weight, BMI and wrist circumference were not evaluated in this study. There have been no reports of differences in nerve size between Japanese and other ethnic groups. In the present study, the cutoff value for symptom onset may be higher in Japanese, and thus, we think additional studies are needed. Second, the cross-sectional area was measured at wrist crease level without bony landmark. Nakamichi et al. reported the ultrasound measurements at three levels, and described that reliable data were obtained [20]. One of the levels was based on wrist crease, and reliable data were obtained even without bony landmarks. On the other hand, many papers measured with bony landmarks (at the level of pisiform). We may need to consider the differences of nerve size in the different levels of median nerve. In addition, automatic image analysis procedures have been reported recently [11]. The results of our manual measurement may need to compare the results with automatic analyzing procedure. Third, the characteristics of patients in which SNAP or CMAP was not derived were not evaluated. Lastly, cut-off values were not confirmed when symptoms appeared on the asymptomatic side. These limitations should be considered in the future study.

## 5. Conclusions

In conclusions, the ultrasonographic and electrophysiological features of unilateral idiopathic CTS patients were evaluated. It was found that the most reliable parameter that reflected clinical symptoms was the distal latency of CMAP. The cut-off values of each parameter are considered to be an index for the onset of the clinical symptoms of CTS. 

## Figures and Tables

**Figure 1 diagnostics-12-02799-f001:**
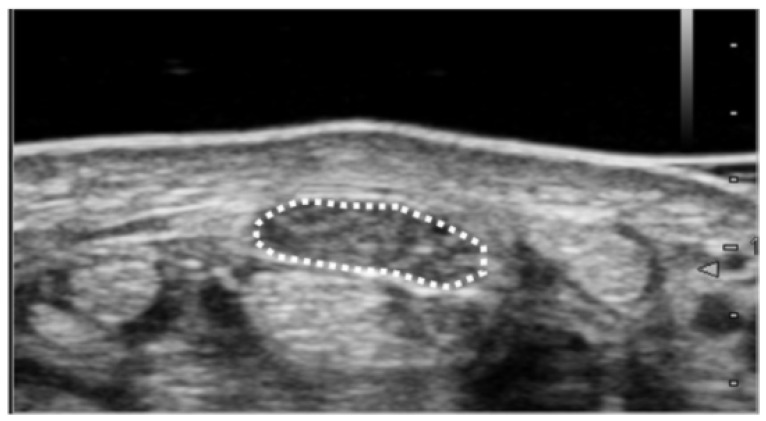
Ultrasound measurement of median nerve cross-sectional area. Cross-sectional area was measured as the only parameter for the image analysis. Measurement of the median nerve cross-sectional area was performed at the wrist crease level. The median nerve was outlined, and its area was calculated.

**Figure 2 diagnostics-12-02799-f002:**
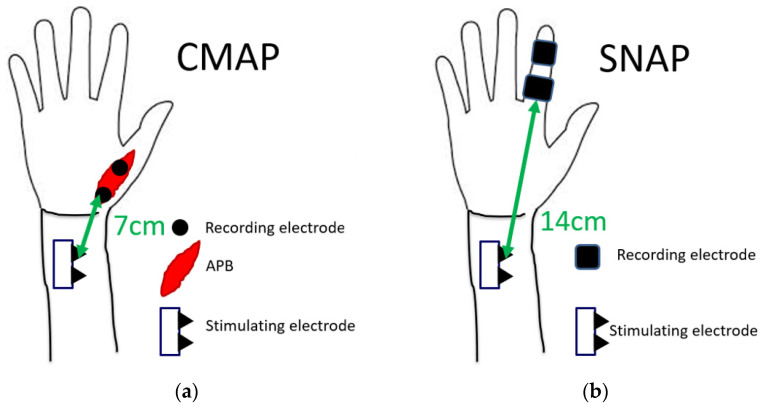
Measurement conditions of latencies of (**a**) CMAP and (**b**) SNAP.

**Figure 3 diagnostics-12-02799-f003:**
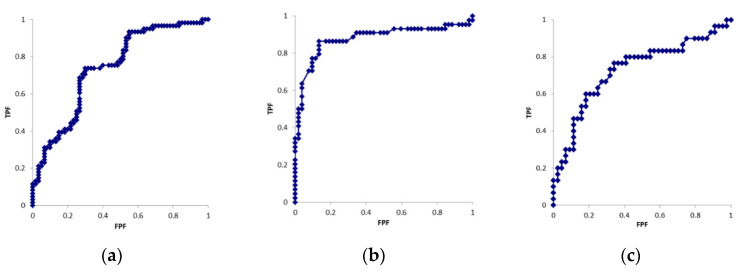
ROC curves of the (**a**) cross-sectional area, (**b**) CMAP latency (**c**) SNAP latency.

**Table 1 diagnostics-12-02799-t001:** Patient demographics.

Gender	Male	22
	Female	39
Age	Ave (SD)	65.3 ± 11.2
	Min	30
	Max	89
Height (cm)	Ave (SD)	157.6 ± 8.6
Weight (kg)	Ave (SD)	61.3 ± 11.9
BMI	Ave (SD)	24.6 ± 3.8
Work Status (%)	Manual work	44.8
	Desk work	13.8
	Unemployed	41.4
Disease period (%)	Less than 3 months	38.6
	3 months to 1 year	33.3
	More than 1 year	28.1
APB strength (MMT, %)	0~1	19.3
	2~3	40.4
	4~5	40.4

**Table 2 diagnostics-12-02799-t002:** Results obtained for each parameter.

	Affected Side	Unaffected Side	AUC	Cut-Off Values
Cross-sectional area (mm^2^)	14.2 ± 4.0	11.3 ± 2.7	0.74	11.9
Distal Latency for CMAP (ms)	6.6 ± 1.8	4.4 ± 0.9	0.88	5.1
Distal Latency for SNAP (ms)	3.5 ± 0.8	2.8 ± 0.4	0.73	3.1

## Data Availability

The datasets analyzed during the present study are available from the corresponding author upon reasonable request.

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
