# Peer review of "Clinical Relevance of Ultrasonographic and Electrophysiological Findings of the Median Nerve in Unilateral Carpal Tunnel Syndrome Patients"

_diagnostics, 2022, doi:10.3390/diagnostics12112799_

Round 1

Reviewer 1 Report

The authors describe diagnostic parameters of CTS in the presence of unilateral symptoms. When comparing the clinically affected side to the clinically not-affected side the authors found significant differences.

1) This is interesting and relevant. Please discuss/clarify the definition and relevance of unilateral CTS-there are multiple studies indicating that CTS ultimately appears in both wrists…

2) Furthermore, most surgeons treat only the symptomatic side at any given time. What is the place for testing in unilateral CTS presentations?

Author Response

The authors describe diagnostic parameters of CTS in the presence of unilateral symptoms. When comparing the clinically affected side to the clinically not-affected side the authors found significant differences.

1) This is interesting and relevant. Please discuss/clarify the definition and relevance of unilateral CTS-there are multiple studies indicating that CTS ultimately appears in both wrists…

Response)In this study, we defined unilateral CTS as patients with characteristic symptoms on one side of the hand and no symptoms on the other side. A single hand surgeon discriminated between the asymptomatic and symptomatic sides based on the interview and clinical findings. Definitely, most of CTS patients have both side symptoms. As it could identify the preclinical condition of the asymptomatic side, it may allow intervention to prevent progression of neuropathy. We added these descriptions in the text. (Page 2, Line 61-63, and Page 2, Line 71-72, and Page 6, Line 139-142)

2) Furthermore, most surgeons treat only the symptomatic side at any given time. What is the place for testing in unilateral CTS presentations?

Response)As described in the response #1, identifying the preclinical condition of the asymptomatic side, it may allow early intervention to avoid surgery. Approximately 25% of unilateral CTS patients went on to develop symptoms of CTS after 6–11 years (Fowler et al. 2011). If we could detect preclinical conditions in the asymptomatic side and define the characteristics of the symptom onset, there is a possibility to prevent the onset of carpal tunnel syndrome or treat in very early stage. (Page 6, Line 139-142)

Reviewer 2 Report

The paper discusses the clinical relevance of ultrasonographic and electrophysiological findings of the median nerve for patients with UCT syndrome. The topic is interesting and important, there is a lot of research on implementation of US image analysis in the support of CTS. The interesting point of this work is implementation of two techniques to assess the CTS. However, the number of issues should be clarified before the paper will be suitable for publication.

1.     The novelty and contribution of this work should be clearly stated in the Introduction

2.     The literature review is poor. The following important papers that investigate this topic should be considered and briefly discussed:

Mallouhi A, Pultzl A, Trieb T, Piza H, Bodner G. Predictors of carpal tunnel syndrome: accuracy of gray-scale and color Doppler sonography. AJR Am J Roentgenol 2006;186:1240–5

Ozsoy-Unubol T, Bahar-Ozdemir Y, Yagci I. Diagnosis and grading of carpal tunnel syndrome with quantitative ultrasound: Is it possible? J Clin Neurosci 2020;75:25–9

Ooi CC, Wong SK, Tan AB, et al. Diagnostic criteria of carpal tunnel syndrome using high-resolution ultrasonography: correlation with nerve conduction studies. Skeletal Radiol 2014;43:1387–94

Obuchowicz R, Kruszynska J, Strzelecki M. Classifying median nerves in carpal tunnel syndrome: Ultrasound image analysis. Biocybernetics Biomed Eng 2021;41:335-351

Fujimoto K, Kanchiku T, Kido K, Imajo Y, Funaba M, Taguchi T. Diagnosis of severe carpal tunnel syndrome using nerve conduction study and ultrasonography. Ultrasound Med Biol 2015;41:2575–80

Tagliafico AS. Peripheral nerve imaging: not only crosssectional area. World J Radiol 2016;8:726–8.

Hirani S. A study to further develop and refine carpal tunnel syndrome (CTS) nerve conduction grading tool. BMC Musculoskeletal Disorders 2019;20(1):581

Chen IJ, Chang KV, Lou YM, Wu WT, Ozcakar L. Can ultrasound imaging be used for the diagnosis of carpal tunnel syndrome in diabetic patients? A systemic review and network meta-analysis. J Neurol 2020;267(7):1887–95

Jablecki CK, Andary MT, Floeter MK, et al. Practice parameter: electrodiagnostic studies in carpal tunnel syndrome. Report of the American Association of Electrodiagnostic Medicine, American Academy of Neurology, and the American Academy of Physical Medicine and Rehabilitation. Neurology 2002;58:1589–92

Werner RA, Andary M. Carpal tunnel syndrome: pathophysiology and clinical neurophysiology. Clin Neurophysiol 2002;113(9):1373–81

Byra M, Hentzen E, Du J, Andre M, Chang EY, Shah S. Assessing the performance of morphologic and echogenic features in median nerve ultrasound for carpal tunnel syndrome diagnosis. J Ultrasound Med 2020;39(6):1165–74 Padua L, Coraci D, Erra C, et al. Carpal tunnel syndrome: clinical features, diagnosis, and management. Lancet Neurol 2016;15:1273–84

3.     What kind of images analysis was performed? If some geometrical parameters were measured, this should be indicated in Fig. 1.

4.     What classifier was used to distinguished between affected and unaffected side parameters?

5.     Obtained results should be compared and discussed with these obtained using automatic image analysis.

Author Response

The paper discusses the clinical relevance of ultrasonographic and electrophysiological findings of the median nerve for patients with UCT syndrome. The topic is interesting and important, there is a lot of research on implementation of US image analysis in the support of CTS. The interesting point of this work is implementation of two techniques to assess the CTS. However, the number of issues should be clarified before the paper will be suitable for publication.

  1. The novelty and contribution of this work should be clearly stated in the Introduction

Response)The contribution of this study is to identify the clinical parameters in which patients perceive their symptoms by comparing the results of clinical examinations of symptomatic and asymptomatic hands. We added the description in the introduction. (Page 2, Line 45-47)

  1. The literature review is poor. The following important papers that investigate this topic should be considered and briefly discussed:

Response)Thank you for your constructive comments. We included these articles in the text.

(Page 7, Line 205-207, Ref 25) Mallouhi A, Pultzl A, Trieb T, Piza H, Bodner G. Predictors of carpal tunnel syndrome: accuracy of gray-scale and color Doppler sonography. AJR Am J Roentgenol 2006;186:1240–5

(Page 6, Line 163, Ref 16) Ozsoy-Unubol T, Bahar-Ozdemir Y, Yagci I. Diagnosis and grading of carpal tunnel syndrome with quantitative ultrasound: Is it possible? J Clin Neurosci 2020;75:25–9

(Page 6, Line 144-146, Ref 8) Ooi CC, Wong SK, Tan AB, et al. Diagnostic criteria of carpal tunnel syndrome using high-resolution ultrasonography: correlation with nerve conduction studies. Skeletal Radiol 2014;43:1387–94

(Page 6, Line 155, Ref 11) Obuchowicz R, Kruszynska J, Strzelecki M. Classifying median nerves in carpal tunnel syndrome: Ultrasound image analysis. Biocybernetics Biomed Eng 2021;41:335-351

(Page 6, Line 163, Ref 17) Fujimoto K, Kanchiku T, Kido K, Imajo Y, Funaba M, Taguchi T. Diagnosis of severe carpal tunnel syndrome using nerve conduction study and ultrasonography. Ultrasound Med Biol 2015;41:2575–80

(Page 7, Line 207-209, Ref 26) Tagliafico AS. Peripheral nerve imaging: not only crosssectional area. World J Radiol 2016;8:726–8.

(Page 7, Line 223-225, Ref 34) Hirani S. A study to further develop and refine carpal tunnel syndrome (CTS) nerve conduction grading tool. BMC Musculoskeletal Disorders 2019;20(1):581

(Page 6, Line 156-157 Ref 13) Chen IJ, Chang KV, Lou YM, Wu WT, Ozcakar L. Can ultrasound imaging be used for the diagnosis of carpal tunnel syndrome in diabetic patients? A systemic review and network meta-analysis. J Neurol 2020;267(7):1887–95

(Page 2, Line 67, Ref 6) Jablecki CK, Andary MT, Floeter MK, et al. Practice parameter: electrodiagnostic studies in carpal tunnel syndrome. Report of the American Association of Electrodiagnostic Medicine, American Academy of Neurology, and the American Academy of Physical Medicine and Rehabilitation. Neurology 2002;58:1589–92

(Page 7, Line 216, Ref 32) Werner RA, Andary M. Carpal tunnel syndrome: pathophysiology and clinical neurophysiology. Clin Neurophysiol 2002;113(9):1373–81

(Page 6, Line 156-157, Ref 14) Byra M, Hentzen E, Du J, Andre M, Chang EY, Shah S. Assessing the performance of morphologic and echogenic features in median nerve ultrasound for carpal tunnel syndrome diagnosis. J Ultrasound Med 2020;39(6):1165–74

(Page 6, Line 163-165, Ref 18) Padua L, Coraci D, Erra C, et al. Carpal tunnel syndrome: clinical features, diagnosis, and management. Lancet Neurol 2016;15:1273–84

  1. What kind of images analysis was performed? If some geometrical parameters were measured, this should be indicated in Fig. 1.

Response)In this study, we only measured the median nerve cross-sectional area. As it was most often reported and reliable imaging parameters for the diagnosis of the CTS. We added the descriptions of the image analysis parameter in the legends of Figure 1. (Page 3, Line 87-90, Figure 1)

  1. What classifier was used to distinguished between affected and unaffected side parameters?

Response)Patients were asked to complete the JSSH version of the CTS instrument. Then, one hand surgeon distinguished between affected and unaffected side by interview and physical examination. We added these descriptions in the text. (Page 2, Line 57-58, and Page 2, Line 71-72, Ref 5)

  1. Obtained results should be compared and discussed with these obtained using automatic image analysis.

Response)We didn’t have that kind of equipment in our institution. We measured the image parameters manually. This may be one of the limitations. We also added the descriptions for the future perspectives of comparing with the automatic image analysis procedures. (Page 8, Line 270-271, Ref 11)

Round 2

Reviewer 2 Report

Thank you for addressing all comments raised in my review. The paper now is suitable for publication.